# Trends in suicide rates by race and ethnicity among members of the United States Army

Lisa A. Brenner[1,2]*, Jeri E. Forster[1,2], Colin G. Walsh[3], Kelly A. Stearns-Yoder[1,2], Mary Jo Larson[4], Trisha A. Hostetter[1], Claire A. Hoffmire[1,2], Jaimie L. Gradus[5], Rachel Sayko Adams[1,4,6]

1 VHA Rocky Mountain Mental Illness Research Education and Clinical Center, Aurora, Colorado, United States of America, 2 University of Colorado, Anschutz Medical Campus, Aurora, Colorado, United States of America, 3 Departments of Biomedical Informatics, Medicine, and Psychiatry, Vanderbilt University Medical Center, Nashville, TN, United States of America, 4 Institute for Behavioral Health, The Heller School for Social Policy and Management, Brandeis University, Waltham, Massachusetts, United States of America, 5 Department of Epidemiology, Boston University School of Public Health, Boston, Massachusetts, United States of America, 6 Department of Health Law, Policy and Management, Boston University School of Public Health, Boston, Massachusetts, United States of America

☯ These authors contributed equally to this work.
* lisa.2.brenner@cuanschutz.edu

**Data Availability Statement:** As this data presented is from the Departments of Defense (DoD) and Veterans Affairs (VA), the investigators

## Abstract

Efforts were focused on identifying differences in suicide rates and time-dependent hazard rate trends, overall and within age groups, by race and ethnicity among United States Army members who returned from an index deployment (October 2007 to September 2014). This retrospective cohort study was conducted using an existing longitudinal database, the Substance Use and Psychological Injury Combat Study (SUPIC). Demographic (e.g., race and ethnicity) and military data from the Department of Defense compiled within SUPIC, as well as Department of Veterans Affairs data were linked with National Death Index records (through 2018) to identify deaths by suicide including those that occurred after military service. The cohort included 860,930 Army Service members (Active Duty, National Guard, and Reserve). Age-adjusted (using the direct standardization method) and age-specific suicide rates per 100,000 person years were calculated and rate ratios (RR) were used for comparisons. Trends were evaluated using hazard rates over time since the end of individuals' index deployments. Among those aged 18–29 at the end of their index deployment, the suicide rate for American Indian/Alaskan Native (AI/AN) individuals was 1.51 times higher (95% confidence interval [CI]: 1.03, 2.14) compared to White non-Hispanic individuals (WNH), and lower for Hispanic and Black non-Hispanic (BNH) than for WNH individuals (RR = 0.65 [95% CI: 0.55, 0.77] and RR = 0.71 [95% CI: 0.61, 0.82], respectively). However, analyses revealed increasing trends in hazard rates post-deployment ($\leq$ 6.5 years) within groups of Hispanic and BNH individuals (Average Annual Percent Change [APC]: 12.1% [95% CI: 1.3%, 24.1%] and 11.4% [95% CI: 6.9%, 16.0%], respectively) with a smaller, increase for WNH individuals (APC: 3.1%; 95% CI: 0.1%, 6.1%). Findings highlight key subgroups at risk for post-deployment suicide (i.e., WNH, AI/AN and younger individuals), as well as heterogeneous trends overtime, with rates and trends varying within race and ethnic groups by age groups. Post-deployment

needed to obtain multiple permissions prior to use. Based on existing DoD and VA policies this data has been deemed sensitive and as such the investigators are not authorized to shared data. Queries can be made to VHAECHMIRECCAdmin@va.gov.

**Funding:** This study was funded by the National Institute of Mental Health (NIH) and Office of the Director at NIH (R01MH120122). Funding to support cohort development was from the National Center for Complementary and Integrative Health (NCCIH; R01 AT008404) and the National Institute on Drug Abuse (NIDA; R01 DA030150). Our Department of Defense data sponsor is Major Ryan C. Costantino, PharmD - formerly it was Dr. Chester Buckenmaier, III until recent retirement. All authors received funding. The funders had no role in study design, data collection and analysis, decision to publish, or preparation of the manuscript. https://www.nimh.nih.gov/ https://www.nih.gov/ https://www.nccih.nih.gov/.

**Competing interests:** The authors have declared that no competing interests exist.

suicide prevention efforts that address culturally relevant factors and social determinants of health associated with health inequities are needed.

## Introduction

Since the conflicts in Iraq and Afghanistan, increased rates of suicide have been noted among military members, and, to date, rates among military members and veterans have been higher than those among civilian cohorts [1, 2]. Across branches of the military, particularly high rates have been identified among members of the Army, who provided substantial ground support during the conflicts [1]. Per the 2019 Department of Defense (DoD) Suicide Event Report, Soldiers had the highest suicide rate of any Service (29.8 per 100,000 persons) [3]. Moreover, findings from the most recent Veterans Health Administration's (VHA) annual suicide report suggested that the 2019 age- and gender-adjusted suicide rate among veterans remained over 50% higher than United States adult civilians, at 26.9 veterans per 100,000 [2]. Considering such findings, efforts have been undertaken to identify significant predictors of risk for suicide-related behaviors among service members and veterans, with the goal of preventing negative outcomes. Whereas some of the predictors are unique to military cohorts (e.g., high levels of combat exposure) [4], others seem to be consistent with those identified among members of the general population (e.g., high levels of suicide among American Indian/Alaska Native [AI/AN] individuals) [5–7].

Recently, increased efforts have focused on identifying trends in suicide rates by subgroups, including by race and ethnicity [8, 9], to help guide tailored prevention efforts. Against the backdrop of findings which suggested a decrease in the overall general United States population 2019 suicide rate, Ramchand et al. conducted a cross-sectional study to examine trends (1999–2019) in suicide rates among United States racial and ethnic groups [8]. Whereas decreases between 2018 and 2019 were noted among White and AI/AN individuals, age adjusted rates increased for Black and Asian American or Pacific Islander (AAPI) individuals [8]. Moreover, the increasing trend for these two groups dated back to 2014. Between 2014 and 2019, the suicide rate for Black and AAPI individuals increased by 30% and 16%, respectively [8]. Without such analyses, important information regarding changes (i.e., trends) by subgroup can be masked, thereby precluding efforts to develop interventions aimed at addressing culturally relevant factors (e.g., racism) known to increase risk for a wide-range of negative health outcomes [10, 11].

A recent analysis undertaken to explore trends among AI/AN veterans provides further support for these assertions. Mohatt et al. conducted a retrospective cohort analysis of AI/AN individuals who received care within the Veterans Health Administration between 2002 and 2014 and found that the age-adjusted suicide rates more than doubled over the 15-year observation period (19.1–47.0/100,000 Person Years) [9]. Also of note, was the finding that the youngest age group (18–39) exhibited the highest risk [9]. Taken together, these findings suggest that investigation into suicide rates among racialized minorities with current or past military service should consider if rates and trends vary within subgroups by age group.

Nonetheless, large scale efforts to examine trends in suicide rates over time among United States military personnel by race and ethnic groups have been limited. Toward this end, Departments of Defense (DoD) and Veterans Affairs (VA) data were merged with National Death Index (NDI) records to explore suicide rates and trends by race and ethnicity among United States Army members. Individuals in the cohort returned from a deployment during

the middle years of the conflicts in Iraq and Afghanistan. The goal of this effort was to high-light potential and actionable between group differences in suicide risk based on race and ethnicity and age.

## Materials and methods

### Data sources

As data (which were secondary in nature) were collected from multiple existing sources, permissions to not obtain written consent were received from all pertinent institutions associated with the DoD, VA (e.g., Colorado Multiple Institutional Review Board), and investigators' academic affiliates.

**Substance Use and Psychological Injury Combat Study (SUPIC).** SUPIC is a longitudinal database that Brandeis University researchers developed which includes a total of N = 865,640 individuals comprised of Army Active Duty (AD), National Guard (NG), and Reserve (RS) members returning from an Operation Enduring Freedom (OEF)/Operation Iraqi Freedom (OIF)/Operation New Dawn (OND) deployment between Fiscal Years 2008–2014 (i.e., October 1, 2007 –September 30, 2014 [12]. Service members were followed longitudinally from the end of their first deployment during this study window (referred to throughout as the study 'index deployment'), though some had deployed prior to and some deployed subsequent to the index deployment. Deployment data (index and prior history) were obtained from the Contingency Tracking System. Demographic characteristics were drawn from the DoD's Defense Enrollment Eligibility Records System.

**VA/DoD Mortality Data Repository (VA/DoD MDR) [13].** The VA/DoD MDR contains all-cause mortality data from the NDI, the gold standard for capturing cause and date of death [14].

### Measures

**Study cohort.** The analytic cohort of 860,930 (99.5% of the original file) was determined using the following criteria. All original records, excluding 141 without a usable Social Security Number (SSN), were searched in the NDI data and then merged with the SUPIC data. Any record where death occurred prior to the end of the index deployment was removed (N = 1,123). Other inclusion criteria required were: 1) presence of data for military component at the end of the index deployment; 2) index deployment ≤ 5 years; and 3) for those records with a match in the VHA medical record data, we required SSN, date of birth and gender consistency. These final three criteria resulted in the removal of N = 3,446 records.

**Demographics.** Data obtained from SUPIC data files included: age at the end of the index deployment, gender, race and ethnicity (White Not Hispanic [WNH], Black Not Hispanic [BNH], Hispanic, AAPI, AI/AN, and other/unknown); military-related information (e.g., rank); and, Fiscal Year of return from index deployment.

**Death by suicide.** Data through 2018 were obtained from the VA/DoD MDR [13]. Death by suicide was determined by identifying NDI records containing ICD-10 codes X60-X84 and Y87.0 as the underlying cause of death.

### Statistical analyses

Demographic and military characteristics were summarized for the overall base cohort and by component (i.e., AD, NG and RS). Additionally, these same variables were summarized by deployment group (first deployers [i.e., the index deployment was the first deployment] and

2 + deployers) and component. All analyses were performed in SAS software v9.4 (SAS Institute, Cary, NC), R v4.1.1 [15], and Joinpoint Regression Program v4.9.0.1 [16, 17].

Crude and age-adjusted (using the direct standardization method) suicide rates were calculated per 100,000 person years over the time period October 1, 2007 to December 31, 2018 by race and ethnicity (WNH, BNH, Hispanic, AAPI, AI/AN, other, unknown). Available follow-up time was thus up to 11 years post index deployment. Age-adjusted rates were standardized based on the 2000 U.S. population [18], using age categories 18–24, 25–29, 30–34, and 35+. As more fine-grained age groups result in better adjustment, the smallest feasible age categories were chosen for age-adjustment (i.e., each age/racial or ethnic group combination must have had no fewer than 5 events). Age categories for age-specific rates were calculated to maximize reporting. Rates with cell sizes <16 were denoted as unreliable, and <10 were suppressed [19]. All rates for the unknown race group were suppressed. Rate ratios were used to compare age-adjusted and age-specific rates. All rates are presented with 95% confidence intervals (CIs). As suicide is a rare event, crude rates were presented with exact CIs, age-adjusted rates are presented with CIs based on the gamma distribution [20], and rate ratios with CIs based on the inverse of the $F$ distribution [21].

Hazard rates for suicide, over time since the end of index deployment were calculated by race and ethnicity using the life-table method. Rates were presented per 100,000 alive at the beginning of each 1-year interval for years 0–8 post-deployment, and the 3-year interval for years 8–11 post-deployment (due to small cell sizes). Average annual percent change (APC) in hazard rates were estimated using linear, segmented trend analysis of the estimated rates [16]. No group could have more than two different segments (slopes) given the number of years of data available, and each group's best fit model was based on weighted Bayesian Information Criterion. Pairwise tests of parallelism were then used to compare trends between racial/ethnic groups [22]. This analysis finds a common best fit model for the comparison groups and the statistical test is then performed to determine if the trends differ. Due to small cell sizes, hazard rates, APC, and trend comparisons for AI/AN, other, and unknown groups were not calculated.

## Results

Most individuals in the overall sample and across components were WNH (>58%), followed by BNH (12.7%-18%), Hispanic (7.7%-12.4%), AAPI (2.9%-10.3%), and AI/AN (0.9%-1.0%). The overall cohort was mostly 18–29 years of age at the end of their index deployment (62.4%), as were those identified as being AD (67.7%). Lower prevalence of this age group was noted among those in the NG (53.6%) and RS (48.2%). See Table 1 for demographic and military characteristics. As expected, first deployers were comprised of mostly Junior Enlisted members (61.2%) whereas only 18.0% of members who deployed two or more times were Junior Enlisted. As such, only 26.9% of first deployers were Senior Enlisted/Warrant Officers, while 67.9% of multiple deployers were in this rank group. Deployment group patterns were consistent across components, and this is also reflected in the distribution of age across deployment groups (higher prevalence of younger members in AD). For additional data regarding variables summarized by deployment group (first deployers and 2+ deployers) and component, see S1 Table.

Adjusting for age, suicide rates (per 100,000 person years) by race and ethnicity were: WNH—32.75 (95% confidence interval [CI]: 30.53, 35.12); BNH—12.93 (95% CI: 11.03, 15.21), Hispanic—18.88 (95% CI: 14.92, 23.82); AAPI—25.22 (95% CI: 19.98, 31.82); and, AI/AN– 49.28 (95% CI: 29.14, 80.88). See Table 2.

Upon examination of age-specific rates, among those 18–29, AI/AN individuals had a suicide rate that was 1.51 (95% CI: 1.03, 2.14) times higher than for WNH individuals.

**Table 1. Sample characteristics overall and by component.**

| | Overall (N = 860,930) | Active Duty (N = 573,531) | National Guard (N = 206,332) | Reserve (N = 81,067) |
|---|---|---|---|---|
| **Age Category at End of Index Deployment** | | | | |
| 18–24 | 320,548 (37.2%) | 235,117 (41.0%) | 64,651 (31.3%) | 20,780 (25.6%) |
| 25–29 | 217,275 (25.2%) | 152,918 (26.7%) | 46,014 (22.3%) | 18,343 (22.6%) |
| 30–34 | 117,585 (13.7%) | 79,606 (13.9%) | 27,686 (13.4%) | 10,293 (12.7%) |
| 35–39 | 92,002 (10.7%) | 58,442 (10.2%) | 24,440 (11.8%) | 9120 (11.3%) |
| 40+ | 113,520 (13.2%) | 47,448 (8.3%) | 43,541 (21.1%) | 22,531 (27.8%) |
| **Gender** | | | | |
| Male | 766,489 (89%) | 512,087 (89.3%) | 186,378 (90.3%) | 68,024 (83.9%) |
| Female | 94,441 (11%) | 61,444 (10.7%) | 19,954 (9.7%) | 13,043 (16.1%) |
| **Race/Ethnicity** | | | | |
| American Indian/Alaskan Native | 7918 (0.9%) | 5186 (0.9%) | 1989 (1.0%) | 743 (0.9%) |
| Asian or Pacific Islander | 68,699 (8.0%) | 58,887 (10.3%) | 5877 (2.9%) | 3935 (4.9%) |
| Black non-Hispanic | 143,350 (16.6%) | 103,092 (18.0%) | 26,160 (12.7%) | 14,098 (17.4%) |
| White non-Hispanic | 539,434 (62.7%) | 333,404 (58.1%) | 154,338 (74.8%) | 51,692 (63.8%) |
| Hispanic | 91,365 (10.6%) | 65,475 (11.4%) | 15,832 (7.7%) | 10,058 (12.4%) |
| Other | 7838 (0.9%) | 6203 (1.1%) | 1371 (0.7%) | 264 (0.3%) |
| Unknown/Missing | 2326 (0.3%) | 1284 (0.2%) | 765 (0.4%) | 277 (0.3%) |
| **Index Deployment was** | | | | |
| First Deployment | 598,335 (69.5%) | 386,283 (67.4%) | 152,539 (73.9%) | 59,513 (73.4%) |
| Second or Higher Deployment | 262,595 (30.5%) | 187,248 (32.6%) | 53,793 (26.1%) | 21,554 (26.6%) |
| **Rank Group** | | | | |
| Junior Enlisted | 413,463 (48.0%) | 291,274 (50.8%) | 94,495 (45.8%) | 27,694 (34.2%) |
| Senior Enlisted/Warrant Officer | 339,205 (39.4%) | 196,614 (34.3%) | 86,609 (42.0%) | 36,108 (44.5%) |
| Junior Officer | 70,200 (8.2%) | 49,424 (8.6%) | 13,596 (6.6%) | 7180 (8.9%) |
| Senior Officer | 38,057 (4.4%) | 22,817 (4.0%) | 6879 (3.3%) | 8361 (10.3%) |
| Missing | 5 (0%) | 2 (0%) | 2 (0%) | 1 (0%) |
| **Fiscal Year of Return from Index Deployment** | | | | |
| 2008–09 | 316,420 (36.8%) | 223,354 (38.9%) | 67,808 (32.9%) | 25,258 (31.2%) |
| 2010–11 | 326,101 (37.9%) | 209,520 (36.5%) | 83,977 (40.7%) | 32,604 (40.2%) |
| 2012–14 | 218,409 (25.4%) | 140,657 (24.5%) | 54,547 (26.4%) | 23,205 (28.6%) |

Additionally, both Hispanic and BNH members had significantly lower rates than WNH individuals. The protective pattern for Hispanic and BNH individuals compared to White non-Hispanic individuals was also present among those 30+ years of age. Age-specific suicide rates by race and ethnicity are provided in Table 3.

Hazard rates and trends for suicide estimated over time since index deployment are provided in Table 4, Fig 1, and S2 Table. Trend analysis revealed that WNH, BNH, and Hispanic military members all had best fitting trend lines with one joint where suicide hazards increased initially post-deployment and subsequently declined. Conversely, the best fitting trend line for the AAPI group had zero joints (i.e., singly linear trend). Overall, lines (and therefore most estimated hazard rates) for WNH and AAPI individuals were consistently higher than those for BNH and Hispanic members. Not all best fitting trends demonstrated statistically significant increases or decreases, however. Specifically, only BNH military members had a significant average annual percent increase in hazard rates of 11.4% (95% CI: 6.9%, 16.0%) between the end of the index deployment and 3.5 years post-deployment followed by a significant

**Table 2. Suicide rates per 100,000 person years (October 1, 2007- December 31, 2018).**

|  | Crude Rate (95% CI) | Age-adjusted Rate* (95% CI) | Rate Ratio (95% CI) |
|---|---|---|---|
| Race /Ethnicity |  |  |  |
| American Indian/Alaskan Native | 65.41 | 49.28 | 1.50 |
|  | (47.53, 87.82) | (29.14, 80.88) | (0.88, 2.49) |
| Asian or Pacific Islander | 41.14 | 25.22 | 0.77 |
|  | (36.17, 46.59) | (19.98, 31.82) | (0.60, 0.98) |
| Black non-Hispanic | 23.18 | 12.93 | 0.39 |
|  | (20.53, 26.08) | (11.03, 15.21) | (0.33, 0.47) |
| White non-Hispanic | 42.08 | 32.75 | Ref |
|  | (40.21, 44.02) | (30.53, 35.12) |  |
| Hispanic | 27.30 | 18.88 | 0.58 |
|  | (23.71, 31.27) | (14.92, 23.82) | (0.45, 0.74) |
| Other | 26.40 | Suppressed | Suppressed |
|  | (15.65, 41.72) |  |  |

*Age-adjusted rates based on the following age categories: 18–24, 25–29, 30–34, 35+

**Table 3. Age-specific suicide rates and rate ratios.**

|  | Rate (95% CI) | Rate Ratio (95% CI) |
|---|---|---|
| **18–29** |  |  |
| American Indian/Alaskan Native | 73.80 | 1.51 |
|  | (50.48, 104.18) | (1.03, 2.14) |
| Asian or Pacific Islander | 48.91 | 1.00 |
|  | (42.39, 56.16) | (0.86, 1.16) |
| Black Non-Hispanic | 34.66 | 0.71 |
|  | (30.26, 39.52) | (0.61, 0.82) |
| White Non- Hispanic | 48.94 | Ref |
|  | (46.40, 51.59) |  |
| Hispanic | 31.88 | 0.65 |
|  | (27.08, 37.30) | (0.55, 0.77) |
| Other | 65.57* | 1.34 |
|  | (34.91, 112.13) | (0.71, 2.30) |
| **30+** |  |  |
| American Indian/Alaskan Native | 50.21* | 1.65 |
|  | (25.94, 87.71) | (0.85, 2.91) |
| Asian or Pacific Islander | 24.48 | 0.80 |
|  | (17.99, 32.56) | (0.58, 1.09) |
| Black Non-Hispanic | 9.79 | 0.32 |
|  | (7.35, 12.77) | (0.24, 0.43) |
| White Non-Hispanic | 30.45 | Ref |
|  | (27.86, 33.22) |  |
| Hispanic | 19.07 | 0.63 |
|  | (14.24, 25.00) | (0.46, 0.83) |
| Other | Suppressed | Suppressed |

*Unreliable, N<16

**Table 4. Estimated Annual Percent Change (APC) and change timepoint (if applicable) by trend segment.**

| Race/Ethnicity | Segment Number | Change Timepoint | APC (95% CI) | Test for Parallelism p-value |
|---|---|---|---|---|
| Asian or Pacific Islander | 1 | - | 0.6 (-3.0, 4.4) | Asian or Pacific Islander vs Black non-Hispanic: 0.07 |
| Black non-Hispanic | 1 | 3.5 years | **11.4 (6.9, 16.0)** | Asian or Pacific Islander vs White non-Hispanic: 0.12 |
| | 2 | | **-6.4 (-7.9, -5.0)** | Asian or Pacific Islander vs Hispanic: 0.03 |
| White non-Hispanic | 1 | 5.5 years | **3.1 (0.1, 6.1)** | Black non-Hispanic vs White non-Hispanic: 0.047 |
| | 2 | | -2.0 (-6.2, 2.4) | Black non-Hispanic vs Hispanic: 0.002 |
| Hispanic | 1 | 6.5 years | **12.1 (1.3, 24.1)** | White non-Hispanic vs Hispanic: 0.01 |
| | 2 | | -14.0 (-37.1, 17.4) | |

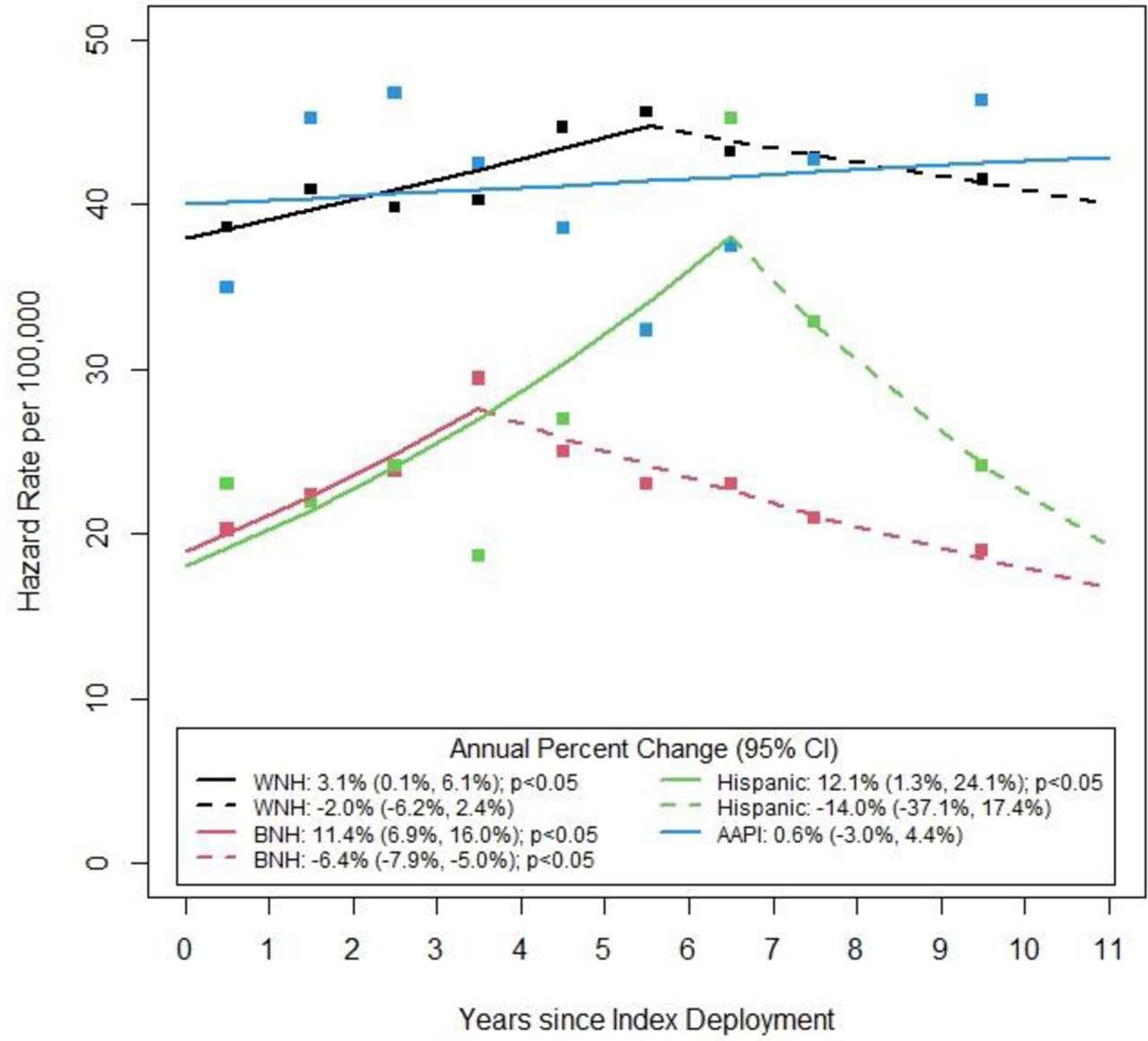

**Fig 1. Hazard rates per 100,00 alive at beginning of the interval with trend lines.**

decreasing trend between 3.5- and 11-years post-deployment (-6.4%, 95% C: -7.9%, -5.0%). While WNH military members also had a significant average annual percent increase in hazard rates of 3.1% (95% CI: 0.1%, 6.1%) between the end of the index deployment and 5.5 years post-deployment, the decline observed in later years for this group was -2.0% (95% CI: -6.2%, 2.4%). Like the WNH group, Hispanic military members had an initial significant average annual percent increase in hazard rates of 12.1% (95% CI: 1.3%, 24.1%) between the end of the index deployment and 5.5 years post-deployment followed by a large but non-significant decline in hazard rates. Finally, AAPI service members experienced relatively stable hazard rates for suicide over the 11 years of follow-up post-index deployment (APC = 0.6%; 95% CI: -3.0%, 4.4%). The pairwise tests for parallelism revealed significant differences in the common model trends (i.e., we reject that common model slopes are parallel) between AAPI and Hispanic members (p = 0.03), BNH and WNH members (p = 0.047), BNH and Hispanic members (p = 0.002), and, WNH and Hispanic members (p = 0.01). Significant trend differences were not observed between WNH and AAPI members, or BNH and AAPI members. See Table 4 and Fig 1.

## Discussion

Findings highlight differences in rates and time trends (October 2007 –December 2018) for race and ethnic subgroups. Consistent with literature regarding the general population, age-adjusted suicide rates varied by subgroups and ranged from 49.28/100,000 (AI/AN individuals) to 12.93/100,000 (BNH individuals). Higher identified overall age-adjusted rates among WNH (32.75/100,000) and AI/AN individuals are consistent with previous rates noted among these groups in both military and veteran cohorts [2, 3, 5, 6, 9].

However, analysis by race/ethnic subgroup alone is not sufficient to guide prevention efforts; age group differences also must be taken into account. For example, whereas the age-specific suicide rate for BNH members over the age of 30 was 9.79 per 100,000, the rate for BNH individuals 29 and under was 34.66 per 100,000. Similarly, even though rates among AI/AN individuals were the highest overall, rates were notably different between younger (18–29 years 73.80/100,000) and older (30+ years 50.21/100,000) individuals. These findings suggest that for some groups, risk factors (e.g., race and ethnicity, age) may interact in a manner which requires further attention. Moreover, these findings are in-line with previous results among military cohorts suggesting that younger individuals, overall, are at greater risk for suicide [23].

Data also suggest that examining overall suicide rates by race and ethnicity, without also looking at time since deployment, might mask important periods of increased risk. As highlighted in Table 4, Fig 1, and S2 Table, trend lines of hazard rates in the years since index deployment differed by race and ethnicity. These differences may highlight unique stressors faced by individual groups post-deployment, as well as opportunities for intervention. Of note, findings from this study focused on years post-deployment versus transition out of the military, with periods of increased risk in the years following deployment observed. For some individuals in the cohort, years post deployment coincided with transition from military to civilian life. Previous work has shown increased risk for death by suicide following a wide range of transitions [24–27]. Warner et al. found three time periods of significantly increased risk for suicide among Soldiers over a 15-month deployment cycle [28]. The first occurred in the second month and was hypothesized to be a result of separation from important social relationships. The second was after six months deployed, which was thought to be related to feelings of isolation, and the third was near the end of the Soldier's deployment and was hypothesized to be related to exposure to stressors at home. The period covered in our study starts a 4[th]

segment of the deployment cycle, the months and years post-deployment. Moreover, in a recent article Ravindran and colleagues examined prevalence, patterns, and associated characteristics of death by suicide post-separation from the military [23]. Findings suggested that risk was highest in the first year and declined "only modestly" over the study period (6 years; p. 1) [23]. In terms of race and ethnicity, Black individuals were at decreased risk compared to White individuals, and Hispanic individuals were at decreased risk compared to non-Hispanic individuals. The team also found that in comparison to those who left service at an older age (≥ 40 years) individuals aged 17–19 and 25–29 had suicide hazard rates which were 4.5 and 2.6 times higher, respectively. Without additional analyses regarding age-specific suicide rates by race and ethnicity, results regarding risk among Black and Hispanic individuals should be interpreted with caution.

Policy makers, clinicians, and researchers are encouraged to focus suicide prevention efforts on specific groups and times of elevated risk, with the goal of designing culturally appropriate community- and individual-based interventions. This elevated risk continues beyond the time of military service, and may be especially acute among service members who do not link to the VHA for their medical care. At the same time, we encourage readers to conceptualize differences in suicide outcomes in the context of systemic discrimination, social and structural processes that create inequality [29]. For example, in their recent article regarding challenges and strategies to address conceptualization, contextualization, and operationalization of race in quantitative health research, Lett et al. highlight specific structural factors that may influence inequity among minoritized individuals including "cultural norms, policies, laws, and practices" (p. 158) [29]. For example, according to a recent DoD report, the Armed Forces has a long history of segregation, which persisted until after the Korean War (1954). Inequities remain. For example, in 2020, minority members filled half of the lowest rank positions (Enlisted 1-Enlisted 2) and only 10% of the highest rank positions (Officer 9-Offficer 10) [30]. As such, exploration of suicide-prevention interventions aimed at addressing social determinants of health (e.g., economic stability, social and community contexts) and reducing inequities are paramount [31].

One limitation of this study was that self-reported choices in military records for race and ethnicity were constrained to a single option with no category for multiracial/mixed race. As such, limitations include misclassification error introduced by collapsing groups who may have different lived experiences of systemic discrimination [29]. Addressing issues related to measurement of race and ethnicity will be critical to ensuring that studies not only identify disparities, but also inform tailored approaches to reducing disparities in those affected. For some cohorts, small group sizes posed a range of challenges. For example, for some groups we were not able to present statistical findings and/or further adjust rates (e.g., gender adjustment). While this was necessary to avoid presenting unreliable results and to preserve confidentiality, it limited our ability to make all comparisons of interest and examine suicide trends for all groups (e.g., AI/AN individuals). Smaller samples for some racial groups (e.g., AAPI individuals) also contributed to reduced power for statistical comparisons and resulted in some imprecisely measured estimates, as evidenced by the width of the confidence intervals, and should be interpreted with this in mind. Finally, some level of uncertainty always exists in terms of cause of death reported in NDI data.

Data presented support the importance of looking at large data sets over time, with the goal of identifying trends in suicide by subgroups. Such work can only be completed using longitudinal datasets, and like this study may require merging of data from separate sources. Combining such data has allowed for the identification of differences in subgroup trends over time. In response, suicide prevention efforts must address systemic factors known to contribute to health inequities among minoritized groups.

Findings highlight specific racial/ethnic (e.g., AI/AN individuals) and age (less than 30 years) groups, as well as time periods (e.g., up to 5 years post-deployment) of increased suicide risk. Community and individual suicide prevention efforts aimed at addressing both culturally relevant factors, as well as social determinants of health are indicated.

## Supporting information

**S1 Table. Sample characteristics by first deployers and 2+ deployers within component.**
(DOCX)

**S2 Table. Hazard rates by years since end of index deployment.**
(DOCX)

## Author Contributions

**Conceptualization:** Lisa A. Brenner, Jeri E. Forster, Colin G. Walsh, Kelly A. Stearns-Yoder, Mary Jo Larson, Claire A. Hoffmire, Jaimie L. Gradus, Rachel Sayko Adams.

**Data curation:** Lisa A. Brenner, Jeri E. Forster, Mary Jo Larson, Trisha A. Hostetter, Claire A. Hoffmire, Jaimie L. Gradus, Rachel Sayko Adams.

**Formal analysis:** Jeri E. Forster, Claire A. Hoffmire, Jaimie L. Gradus, Rachel Sayko Adams.

**Funding acquisition:** Lisa A. Brenner, Jeri E. Forster, Colin G. Walsh, Mary Jo Larson, Rachel Sayko Adams.

**Investigation:** Lisa A. Brenner, Jeri E. Forster, Mary Jo Larson, Rachel Sayko Adams.

**Methodology:** Lisa A. Brenner, Jeri E. Forster, Mary Jo Larson, Trisha A. Hostetter, Claire A. Hoffmire, Rachel Sayko Adams.

**Project administration:** Kelly A. Stearns-Yoder, Rachel Sayko Adams.

**Resources:** Lisa A. Brenner, Mary Jo Larson.

**Validation:** Trisha A. Hostetter.

**Visualization:** Jeri E. Forster, Trisha A. Hostetter.

**Writing – original draft:** Lisa A. Brenner, Jeri E. Forster, Colin G. Walsh, Kelly A. Stearns-Yoder, Mary Jo Larson, Trisha A. Hostetter, Claire A. Hoffmire, Jaimie L. Gradus, Rachel Sayko Adams.

**Writing – review & editing:** Lisa A. Brenner, Jeri E. Forster, Colin G. Walsh, Kelly A. Stearns-Yoder, Mary Jo Larson, Trisha A. Hostetter, Claire A. Hoffmire, Jaimie L. Gradus, Rachel Sayko Adams.

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
