## [Editor Report · Decision Letter 0]

1 Aug 2022

PONE-D-22-19049

Trends in Suicide Rates by Race/Ethnicity among Members of the United States Army

PLOS ONE

Dear Dr. Brenner:

Thank you for submitting your manuscript to PLOS ONE. After careful consideration, we feel that it has merit but does not fully meet PLOS ONE’s publication criteria as it currently stands. Therefore, we invite you to submit a revised version of the manuscript that addresses the points raised during the review process.

We have conducted the initial review of criteria for publication, and discovered the following sections that require attention: Financial Disclosure, Ethics Statement and Data Availability. Specifically, these three sections require expansion and where indicated, specific explanations.

The left margin of the manuscript draft provides an overview of information and templates/examples of what is required; additional information may be found at the following links.

https://journals.plos.org/plosone/s/criteria-for-publication

https://journals.plos.org/plosone/s/submission-guidelines#loc-guidelines-for-specific-study-types
https://journals.plos.org/plosone/s/criteria-for-publication#loc-7

https://journals.plos.org/plosone/s/data-availability We look forward to the revision of these sections so we may continue processing the manuscript.

Please submit your revised manuscript by August 9, 2022.  If you will need more time than this to complete your revisions, please reply to this message or contact the journal office at plosone@plos.org. Please include the following items when submitting your revised manuscript:A rebuttal letter that responds to each point raised by the academic editor and reviewer(s). You should upload this letter as a separate file labeled 'Response to Reviewers'.A marked-up copy of your manuscript that highlights changes made to the original version. You should upload this as a separate file labeled 'Revised Manuscript with Track Changes'.An unmarked version of your revised paper without tracked changes. You should upload this as a separate file labeled 'Manuscript'. We look forward to receiving your revised manuscript.

Kind regards,

Darrell Eugene Singer, M.D., M.P.H.

Academic Editor

PLOS ONE

Journal Requirements:

" ext-link-type="uri" xlink:type="simple">https://journals.plos.org/plosone/s/file?id=ba62/PLOSOne_formatting_sample_title_authors_affiliations.pdf"

"This study was funded by the National Institute of Mental Health and Office

of the Director at NIH (R01MH120122). Funding to support cohort development was from the

National Center for Complementary and Integrative Health (NCCIH; R01 AT008404) and the

National Institute on Drug Abuse (NIDA; R01 DA030150). Our Department of Defense data

sponsor is Major Ryan C. Costantino, PharmD - formerly it was Dr. Chester Buckenmaier, III

until recent retirement."

"This study was funded by the National Institute of Mental Health (NIH) and Office of the Director at NIH (R01MH120122). Funding to support cohort development was from the National Center for Complementary and Integrative Health (NCCIH; R01 AT008404) and the National Institute on Drug Abuse (NIDA; R01 DA030150). Our Department of Defense data sponsor is Major Ryan C. Costantino, PharmD - formerly it was Dr. Chester Buckenmaier, III until recent retirement.

All authors received funding.

https://www.nimh.nih.gov/

https://www.nih.gov/

" ext-link-type="uri" xlink:type="simple">https://www.nccih.nih.gov/"

5. Please amend either the title on the online submission form (via Edit Submission) or the title in the manuscript so that they are identical.

Additional Editor Comments:

Dear Dr. Brenner:

My apologies for the delay in responding to your message- (I was experiencing a few technical issues on this end). I am resending the message I'd previously sent with this Minor Revision request- specifically, to address the PLOS ONE's data availability, ethics statement and financial disclosure requirements and details within the disclosures. PLOS won't publish without a deeper explanation of your data concerns.

I have have two reviewers prepared to formally review your manuscript; I have also reviewed it myself and do not anticipate any major problems or concerns (outside of the previously mentioned concerns-).

We look forward to continuing to process the manuscript subsequent to resubmission.

Regards, Darrell Singer

Dr. Brenner:

Thank you for your interest in PLOS ONE and submission of your manuscript "Trends in Suicide Rates by Race and Ethnicity among Members of the United States Army".

We are conducting the initial review of criteria for publication, and discovered the following sections that require attention: Financial Disclosure, Ethics Statement and Data Availability.

Specifically, these three sections require expansion and where indicated, specific explanations. The left margin of the manuscript draft provides an overview of information and templates/examples of what is required; additional information may be found at the following links.

https://journals.plos.org/plosone/s/criteria-for-publication

https://journals.plos.org/plosone/s/submission-guidelines#loc-guidelines-for-specific-study-types

https://journals.plos.org/plosone/s/criteria-for-publication#loc-7

https://journals.plos.org/plosone/s/data-availability

We look forward to the revision of these sections so we may continue processing the manuscript.

Regards,

Darrell Eugene Singer, M.D., M.P.H.

PLOS ONE
---

## [Author Response · Author response to Decision Letter 0]

23 Aug 2022

Thank you for the feedback provided. We have addressed each change requested as outlined below. 

The manuscript has been significantly revised to meet style requirements. These changes have not been identified (e.g., track changes) in the revised manuscript. Significant wording changes have been noted. 

2. The following information regarding data use was added to the manuscript.

As data (which were secondary in nature) were collected from multiple existing sources, permissions to not obtain written consent were received from all pertinent institutions associated with the Department of Defense, Department of Veterans Affairs (e.g., Colorado Multiple Institutional Review Board), and investigators’ academic affiliates.

This change has been made.

These have been added

5. Please review your reference list to ensure that it is complete and correct. 

The references have been re-checked and revised.

6. Data sharing

As this data presented is from the Departments of Defense (DoD) and Veterans Affairs (VA), the investigators needed to obtain multiple permissions prior to use. Based on existing DoD and VA policies this data has been deemed sensitive and as such the investigators are not authorized to shared data.

---

## [Decision Letter · Decision Letter 1]

4 Nov 2022

PONE-D-22-19049R1

Trends in Suicide Rates by Race and Ethnicity among Members of the United States Army

PLOS ONE

Dear Dr. Brenner,

Thank you for re-submitting your manuscript to PLOS ONE.  Your manuscript presents important information on military post-deployment suicides and in particular, demonstrating changes over time periods may provide insight into risks to specific groups of service members. After careful consideration, we feel that it has merit but does not fully meet PLOS ONE’s publication criteria as it currently stands.  We  invite you to submit a revised version of the manuscript that addresses points raised during the review process

Please submit your revised manuscript by Dec 19 2022 11:59PM.  If you will need more time than this to complete your revisions, please reply to this message or contact the journal office at plosone@plos.org. Please include the following items when submitting your revised manuscript:

A rebuttal letter that responds to each point raised by the academic editor and reviewer(s). You should upload this letter as a separate file labeled 'Response to Reviewers'.A marked-up copy of your manuscript that highlights changes made to the original version. You should upload this as a separate file labeled 'Revised Manuscript with Track Changes'.An unmarked version of your revised paper without tracked changes. You should upload this as a separate file labeled 'Manuscript'If applicable, we recommend that you deposit your laboratory protocols in protocols.io to enhance the reproducibility of your results. Protocols.io assigns your protocol its own identifier (DOI) so that it can be cited independently in the future. For instructions see: https://journals.plos.org/plosone/s/submission-guidelines#loc-laboratory-protocols. Additionally, PLOS ONE offers an option for publishing peer-reviewed Lab Protocol articles, which describe protocols hosted on protocols.io. Read more information on sharing protocols at https://plos.org/protocols?utm_medium=editorial-emailutm_source=authorlettersutm_campaign=protocols.

We look forward to receiving your revised manuscript.

Kind regards,

Darrell Eugene Singer, M.D., M.P.H.

Academic Editor

PLOS ONE

Journal Requirements:

Additional Editor Comments:

Your paper will be reconsidered for publication after you address the comments of the editors and peer reviewers and revise the manuscript as appropriate. This request for a revision is not a guarantee the manuscript will be accepted.  The final decision concerning publication will be based on the quality of your responses and the revised paper.  We encourage your review the entirety of both reviewers' comments (below), however, please respond only to the following specific points:

**4. Have the authors made all data underlying the findings in their manuscript fully available?**

*-We appreciate the amendments to this area made by the authors under the first revision request.  The PLOS staff will subsequently determine any final data sharing requirements. (No response required).*

**6.  Review comments to author(s):**

Abstract:

*-*
*Please address the grammatical error in the first sentence.  Suggest changing “Efforts were focused on identify” to “Efforts were focused on identifying” (Edit or response required)*

*-Please indicate the primary analysis for findings (e.g., Hazard or Rate Ratios) and the confidence intervals for each (Edit or response required)*

Introduction

*-When referring to indigenous American populations, the best practice is to define specific groups when using the terms "American Indian" and "Alaska Native", and/or the abbreviation “AI/AN.”  Example- authors write “American Indian/Alaska Natives [AI/AN]” in the last sentence of the first paragraph.  Recommend amending to 'American Indian/Alaska Natives [AI/AN] populations' (or individuals, persons, service members) as is used in the remainder of the paper (e.g., in the second paragraph of the introduction, "...AI/AN individuals." (Edit or response required)-*

*-The statement referring to “…these findings suggest that investigation into subgroups of racial/ethnic military members/Veterans at elevated risk for suicide...”, suggest amending to (or similar) '…these findings suggest that investigation into suicide rates among racialized minorities in the military should consider if...'.  (Edit or response not required but recommended)*

*- "Veteran" is only capitalized at the beginning of a sentence or specific proper nouns (people, groups, places, etc.).  https://www.legion.org/sites/legion.org/files/legion/publications/Legion-Publication-Style-Guide.pdf  (Edit or response required)*

*- "Department of Defense (DoD)" is spelled out and abbreviated/defined as an acronym twice in the introduction, then spelled out or abbreviated multiple times in the remainder of the manuscript.  Recommend editing for consistency.  (Edit or response required)*

Methods:

*-Consider describing the criteria which were used for age classifications.  (Edit or response not required but recommended)*

*-Consider adjusting for gender as well given the generally higher risk of suicide among men (across populations) compared to women?  If not, consider adding to the discussion section. (Edit or response not required but recommended)  *

*- Recommend adding p-values to Table 1. (Edit or response required)*

*- Recommending adding hazards and mortality-related suicide (Yes/No) by component. (Edit or response not required but recommended)  *

Results:

*-Reviewer 1 states "The American Indian/Alaskan Native cohort showed a higher suicide rate, but the lower 95%CI (1.02, 2.13) is almost 1.0. This group is quite small so I would combine it with the “other” race group."  Out of interest in identifying risks to specific groups, it is understandable to maintain separation; however, discussing the limitation of the group size and subsequent confidence interval is appropriate in the discussion section.  (Edit or response not required but recommended)*

*  *

Discussion:

*-Consider discussing the inability to assess suicidality trends among AI/AN person over time.  (Edit or response not required) *

*-Reviewer #2 states: "Given the authors’ exposure of interest is race, they would benefit from using the Lett et al. article cited in the discussion to help contextualize their findings. Specifically, the authors should say more in the discussion about the potential structural factors that may influence inequities in suicide among racialized minorities in the military."  (Edit or response not required)  *

Reviewers' comments:

Reviewer's Responses to Questions

**Comments to the Author**

1. If the authors have adequately addressed your comments raised in a previous round of review and you feel that this manuscript is now acceptable for publication, you may indicate that here to bypass the “Comments to the Author” section, enter your conflict of interest statement in the “Confidential to Editor” section, and submit your "Accept" recommendation.

Reviewer #1: (No Response)

Reviewer #2: (No Response)

2. Is the manuscript technically sound, and do the data support the conclusions?

Reviewer #1: Yes

Reviewer #2: Yes

3. Has the statistical analysis been performed appropriately and rigorously? 

Reviewer #1: Yes

Reviewer #2: Yes

4. Have the authors made all data underlying the findings in their manuscript fully available?

Reviewer #1: Yes

Reviewer #2: No

5. Is the manuscript presented in an intelligible fashion and written in standard English?

Reviewer #1: Yes

Reviewer #2: Yes

6. Review Comments to the Author

Reviewer #1: In this study, Brenner et al. detail the suicide rate among United States Army members who returned from an index deployment with respect to race/ethnicity and age. It is an important contribution for suicide prevention in this population. I would like to raise some points:

Abstract: Rate ratios and confidence intervals must be present.

Methods: My main concern is the way the methods section is written. Please describe the criteria which were used for age classifications. How were hazards for suicide evaluated? Do the authors have any data on psychiatric disorders, mental health visits, or repeated attempts?

Results: The American Indian/Alaskan Native cohort showed a higher suicide rate, but the lower 95%CI (1.02, 2.13) is almost 1.0. This group is quite small so I would combine it with the “other” race group.

Additional analysis to consider for Table 1:

Add p-values.

Add hazards and mortality-related suicide (Yes/No) by component.

Discussion: Asian, Hispanic, and Black veterans were at decreased risk compared to White veterans, regardless of age. Could you confirm these relationships with the support of previous studies?

Reviewer #2: General comments

This retrospective cohort study used the SUPIC database to identify differences in suicide rates and time-dependent hazard rate trends by race and ethnicity among US Army members after index deployment. The authors found that AI/AN persons ages 18-29 had increased risk for suicide compared to WNH persons.

This is an important area of study, as there are limited data on racial inequities in suicide rates in the military. Overall, the study is well designed, straightforward and presented clearly. I do think there are a few minor changes, particularly in the framing of the discussion, that would strengthen the paper.

Abstract

1. Change “Efforts were focused on identify” to “Efforts were focused on identifying”

Intro

1. Best practice is to not let American Indian/Alaska Native or AI/AN stand alone in the body of the paper. Authors write “American Indian/Alaska Natives” in the last sentence of the first paragraph (I believe this is the only time). Instead, suggest “AI/AN people,” “AI/AN persons,” etc, as is used in the remainder of the paper.

2. The authors write that “…these findings suggest that investigation into subgroups of racial/ethnic military members/Veterans at elevated risk for suicide...” Suggest changing to ““…these findings suggest that investigation into suicide rates among racialized minorities in the military should consider if...”

3. DoD is abbreviated multiple times in the intro

Methods

1. Did authors consider adjusting for gender as well given the generally higher risk of suicide among men (across populations) compared to women?

Results

1. see table/figure comment

Discussion

1. Given the authors’ exposure of interest is race, they would benefit from using the Lett et al. article cited in the discussion to help contextualize their findings. Specifically, the authors should say more in the discussion about the potential structural factors that may influence inequities in suicide among racialized minorities in the military.

Tables/figures

1. Can the authors speak more about the inability to assess suicidality trends among AI/AN person over time? It would be nice to see the gross trends over time even if the numbers aren’t large enough to run analyses. At minimum, it seems like this should be added to the limitations section of the paper.

7. PLOS authors have the option to publish the peer review history of their article (what does this mean?). If published, this will include your full peer review and any attached files.

Reviewer #1: **Yes: **Anwar E. Ahmed

Reviewer #2: No

---

## [Author Response · Author response to Decision Letter 1]

3 Dec 2022

Thank you for your interest in our manuscript entitled, Trends in Suicide Rates by Race and Ethnicity among Members of the United States Army, and the opportunity to improve the manuscript based on the provided reviewer feedback. The reviewer recommendations were greatly appreciated. Below we describe how we addressed specific recommendations (see italics). Edits are also highlighted in the revised manuscript. Please note that when doing our revisions, we discovered a small coding error for the estimation of suicide rates and these numbers have been updated. The changes are small (i.e., all rates increased by less than 1 per 100,000 person years), and all significant rate ratios remain – no conclusions have changed.

Abstract:

-Please address the grammatical error in the first sentence. Suggest changing “Efforts were focused on identify” to “Efforts were focused on identifying” (Edit or response required)

 Thank you for pointing out the error, it has been corrected.

-Please indicate the primary analysis for findings (e.g., Hazard or Rate Ratios) and the confidence intervals for each (Edit or response required)

 The method of analysis and confidence intervals have been added to the Abstract.

-Abstract: Rate ratios and confidence intervals must be present.

 We now present the rate ratios with associated confidence intervals.

Introduction

-When referring to indigenous American populations, the best practice is to define specific groups when using the terms "American Indian" and "Alaska Native", and/or the abbreviation “AI/AN.” Example- authors write “American Indian/Alaska Natives [AI/AN]” in the last sentence of the first paragraph. Recommend amending to 'American Indian/Alaska Natives [AI/AN] populations' (or individuals, persons, service members) as is used in the remainder of the paper (e.g., in the second paragraph of the introduction, "...AI/AN individuals." (Edit or response required)

 We have made this recommended change in the text of the manuscript.

-The statement referring to “…these findings suggest that investigation into subgroups of racial/ethnic military members/Veterans at elevated risk for suicide...”, suggest amending to (or similar) '…these findings suggest that investigation into suicide rates among racialized minorities in the military should consider if...'. (Edit or response not required but recommended)

 This change has been made.

- "Veteran" is only capitalized at the beginning of a sentence or specific proper nouns (people, groups, places, etc.). https://www.legion.org/sites/legion.org/files/legion/publications/Legion-Publication-Style-Guide.pdf (Edit or response required)

 The manuscript has been edited as recommended.

- "Department of Defense (DoD)" is spelled out and abbreviated/defined as an acronym twice in the introduction, then spelled out or abbreviated multiple times in the remainder of the manuscript. Recommend editing for consistency. (Edit or response required)

 This has been addressed within the manuscript. 

Methods:

-Consider describing the criteria which were used for age classifications. (Edit or response not required but recommended)

 As the SUPIC cohort was a younger military population at the end of index deployment, cell sizes necessitated collapsing higher age categories into 35+ for age-adjustment. Additionally, the finer-grained the age categories are when adjusting, the better the adjustment will be. We therefore chose the smallest feasible categories for age-adjustment (no less than 5 events per cell). Regarding the age-specific suicide rates, splitting the cohort into 18-29 and 30+ was the only feasible cut point to allow reporting of rates for most of the racial/ethnic groups (at least 10 events per cell were required to report). We have added language to the manuscript to clarify. 

-Consider adjusting for gender as well given the generally higher risk of suicide among men (across populations) compared to women? If not, consider adding to the discussion section. (Edit or response not required but recommended) 

 As women only make up 11% of the cohort and do have a lower suicide rate, the number of events (suicides) available for gender-adjustment was too small to make this feasible. Language has been added to note this as a limitation.

- Recommend adding p-values to Table 1. (Edit or response required)

 Given the very large sample size, even very small and meaningless differences across groups result in statistical significance (all p-values are 0.0001) and therefore not informative. As such we have not added these. 

- Recommending adding hazards and mortality-related suicide (Yes/No) by component. (Edit or response not required but recommended) 

 This paper is focused on trends specifically by race and ethnicity and as such, comparisons by component are beyond the scope. Moreover, as has been highlighted throughout, sample size and overall low base rates of suicide have precluded further stratification by additional factors (e.g., rank) within this manuscript. Members of this team do have a paper under review in which suicide rates by both rank and component, but not by race and ethnicity are evaluated. 

-Methods: My main concern is the way the methods section is written. Please describe the criteria which were used for age classifications. How were hazards for suicide evaluated?

 Please see the response above regarding age classifications. We have added language to the methods to clarify. Hazard rates were calculated using the life-table method and average annual percent change in hazard rates were determined using trend analysis and tests for parallelism were used to determine if trends over time differed across groups. Evaluation of the best fit model are now described. We have added a citation for the trend analysis and tests for parallelism. These methods are used for cancer surveillance by the National Cancer Institute.

Discussion:

-Consider discussing the inability to assess suicidality trends among AI/AN person over time. (Edit or response not required).

 Text related to challenges related to small sample sizes among some cohorts can be found in the discussion section. 

-Reviewer #2 states: "Given the authors’ exposure of interest is race, they would benefit from using the Lett et al. article cited in the discussion to help contextualize their findings. Specifically, the authors should say more in the discussion about the potential structural factors that may influence inequities in suicide among racialized minorities in the military." (Edit or response not required) 

 The following text has been added to the discussion section: For example, in their recent article regarding challenges and strategies to address conceptualization, contextualization, and operationalization of race in quantitative health research, Lett et al. highlight specific structural factors that may influence inequity among minoritized individuals including “cultural norms, policies, laws, and practices” (p. 158). [28] 

Tables/figures

1. Can the authors speak more about the inability to assess suicidality trends among AI/AN person over time? It would be nice to see the gross trends over time even if the numbers aren’t large enough to run analyses. At minimum, it seems like this should be added to the limitations section of the paper.

 We agree that suicide among AI/AN persons is an extremely important topic. Unfortunately, once we break down the number of events (deaths by suicide) among AI/AN individuals across years, this estimation becomes infeasible. We have added this as a limitation.

---

## [Editor Report · Decision Letter 2]

23 Dec 2022

Trends in Suicide Rates by Race and Ethnicity among Members of the United States Army

PONE-D-22-19049R2

Dear Dr. Brenner,

We are pleased to inform you that your manuscript has been judged scientifically suitable for publication and will be formally accepted for publication once it meets all outstanding technical requirements.

Kind regards,

Darrell Eugene Singer, M.D., M.P.H.

Academic Editor

PLOS ONE

Additional Editor Comments (optional):

Dr. Brenner: Congratulations!  Thanks to you and your co-authors for your efforts, patience and this important contribution.  Happy Holidays to you and yours.  Best regards, DS

---

## [Editor Report · Acceptance letter]

6 Jan 2023

PONE-D-22-19049R2 

Trends in Suicide Rates by Race and Ethnicity among Members of the United States Army 

Dear Dr. Brenner:

I'm pleased to inform you that your manuscript has been deemed suitable for publication in PLOS ONE. Congratulations! Your manuscript is now with our production department. 

Kind regards, 

on behalf of

Dr. Darrell Eugene Singer 

Academic Editor

PLOS ONE